# An entropy-controlled objective chip for reflective confocal microscopy with subdiffraction-limit resolution

Jun He[1,4], Dong Zhao [1,4], Hong Liu [2], Jinghua Teng [2] ✉, Cheng-Wei Qiu [3] ✉ & Kun Huang [1] ✉

Planar diffractive lenses (PDLs) with optimized but disordered structures can focus light beyond the diffraction limit. However, these disordered structures have inevitably destroyed wide-field imaging capability, limiting their applications in microscopy. Here, we introduce information entropy $S$ to evaluate the disorder of an objective chip by using the probability of its structural deviation from standard Fresnel zone plates. Inspired by the theory of entropy change, we predict an equilibrium point $S_0 = 0.5$ to balance wide-field imaging (theoretically evaluated by the Strehl ratio) and subdiffraction-limit focusing. To verify this, a $NA = 0.9$ objective chip with a record-long focal length of 1 mm is designed with $S = 0.535$, which is the nearest to the equilibrium point among all reported PDLs. Consequently, our fabricated chip can focus light with subdiffraction-limit size of $0.44\lambda$ and image fine details with spatial frequencies up to 4000 lp/mm experimentally. These unprecedented performances enable ultracompact reflective confocal microscopy for superresolution imaging.

Since the first microscope was invented in 1595 by a Dutch father-son team Hans and Zacharias Janssen[1], optical objectives have developed rapidly with improved performance in aberration correction, field of view, magnification, and numerical aperture, except its imaging resolution being limited by the diffraction of light[2]. Based on refractive optics[3], traditional objectives need multiple elements with carefully designed curvatures and air intervals for better imaging, thus leading to a bulky volume. The situation of objectives has been sustained for over 400 years until subwavelength-thick metalenses were reported with ultracompact volume in 2016[4]. However, high-aspect-ratio dielectric nanobricks in the metalenses exhibit high efficiency under normal incidence only with a small tolerance of tilting angle[5], resulting in large spatial-frequency details of objects being rejected by the metalenses operating in an imaging mode[6,7]. Consequently, the imaging resolutions of the metalenses and their related scanning confocal

microscopies (SCMs) are diffraction-limited to $0.51\lambda/NA$ ($\lambda$ is the wavelength of light and $NA$ is the numerical aperture of the metalens)[8].

Efforts to enhance the resolving power of objectives have also been made by using planar superoscillation lenses[9–12] and supercritical lenses[13–15] that realize subwavelength focal spots by optimizing the destructive or constructive interference between multiple diffracting beams[7,16–18]. Since only the focusing properties of these PDLs are designed without considering the capability of direct wide-field imaging, superoscillation and supercritical lenses are usually used as optical probes in an SCM[7], where an additional refraction-based objective is mandatorily required to collect the transmitted light through the objects. This leads to the fact that all these PDL-based SCMs must operate in transmission mode and are only valid for objects sitting on a transparent substrate that introduces spherical aberration, which requires collection objectives with coverslip collars for

[1]Department of Optics and Optical Engineering, University of Science and Technology of China, Hefei, Anhui 230026, China. [2]Institute of Materials Research and Engineering, Agency for Science Technology and Research (A*STAR), 2 Fusionopolis Way, #08-03, Innovis, Singapore 138634, Singapore. [3]Department of Electrical and Computer Engineering, National University of Singapore, 4 Engineering Drive 3, Singapore 117576, Singapore. [4]These authors contributed equally: Jun He, Dong Zhao. ✉e-mail: jh-teng@imre.a-star.edu.sg; chengwei.qiu@nus.edu.sg; huangk17@ustc.edu.cn

correction[2]. Without involving these issues, reflective SCMs (RSCM) are, therefore, more popular for noninvasive and in vivo imaging of various specimens[19]. Despite the strong requirements from applications, it is still difficult to demonstrate reflective PDL-based SCMs with better-resolving power than commercial SCMs, due to the lack of high-performance planar objectives that possess dual functionalities of focusing and imaging beyond the diffraction limit.

The challenge to design such a planar objective is twofold. First, designing a planar objective for subdiffraction-limit focusing generally leads to an irregularly distributed phase or amplitude[18]. However, such structural disorder is not preferred in its imaging counterpart, where the analytical phase or amplitude is preferred for the constructive reconstruction of objects[7]. Such a dilemma is a fundamental barrier to demonstrating planar objectives with subdiffraction-limit resolution in both focusing and imaging. Second, the diameter of the planar objectives should be sufficiently large to suppress the diffraction effect of collected light by the planar objectives (when working in the imaging mode) for better image formation. Correspondingly, the number of fine structures in the planar objectives is extremely large because of the short wavelength in the visible spectrum, thereby increasing the technical difficulty in the design and fabrication of planar objectives.

To overcome these challenges, we propose a disorder-controlled objective chip that functionally integrates a binary-phase Fresnel zone plate (FZP) and a weakly perturbed few-ring phase mask into a single ultrathin element. By introducing the concept of information entropy, we theoretically predict that an objective chip with its entropy at $S_0 = 0.5$ can maintain the imaging and superfocusing properties simultaneously. Using deep-ultraviolet (DUV) lithography, the fabricated objective chip experimentally exhibits a focal spot of $0.44\lambda$ (below the Rayleigh criterion of $0.51\lambda/NA = 0.57\lambda$) without strong sidebands and the capability of imaging fine objects with a spatial frequency of 4000 lp/mm. Benefiting from this, an ultracompact reflective SCM is built with an imaging resolution (center-to-center) of 200 nm at $\lambda = 405$ nm and a record-long working distance of 1 mm, superior to the state-of-the-art SCMs.

## Results

### Disorder of objective chip

Since standard FZPs have analytical complex transmission, i.e., $U_{FZP} = A_{FZP}e^{i\varphi_{FZP}}$ (where $A_{FZP}$ and $\varphi_{FZP}$ are the amplitude and phase modulation, respectively), they can realize wide-field but diffraction-limit imaging. For an objective chip, its complex transmission $U_{chip} = A_{chip}e^{i\varphi_{chip}}$ deviates irregularly from that of an FZP, thus creating undesired disorder for imaging purposes. Assuming that the deviations are $\Delta A = A_{chip} - A_{FZP}$ for amplitude and $\Delta\varphi = \varphi_{chip} - \varphi_{FZP}$ for phase, we can rewrite the transmission of the objective chip as $U_{chip} = (A_{FZP} + \Delta A)e^{i(\varphi_{FZP} + \Delta\varphi)} = A_{FZP}e^{i\varphi_{FZP}}(1 + \Delta A/A_{FZP})e^{i\Delta\varphi} = U_{FZP} \cdot U_\Delta$, where the deviated transmission $U_\Delta = (1 + \Delta A/A_{FZP})e^{i\Delta\varphi}$ introduces optical aberration in imaging but offers more degrees of freedom for subdiffraction-limit focusing. Thus, for an arbitrary objective chip, its complex transmission contains a standard imaging part $U_{FZP}$ (i.e., FZP) and an additional aberration part $U_\Delta$. In the design of an objective chip, the item $U_\Delta$ is fundamentally important in building the connection between imaging and focusing.

Considering that $U_\Delta$ is distributed spatially in an irregular way, we first investigate its statistical property of such deviation by defining a dimensionless parameter−deviation probability

$$p_1 = \sum_{m=0}^{M-1} \frac{1}{M} \left| \frac{\int_{r_m}^{r_{m+1}} \Delta \cdot rdr}{\int_{r_m}^{r_{m+1}} \Delta_{max} \cdot rdr} \right| \quad (1)$$

where $\Delta = \Delta A$ with a maximum value of $\Delta_{max} = 1$ for the amplitude deviation, $\Delta = \Delta\varphi$ with a maximum value of $\Delta_{max} = \pi$ for the phase deviation, the objective chip is divided into minimum diffraction

subunits (i.e., zones in the FZP) with their boundaries of $r_m = \sqrt{m\lambda f + (m\lambda)^2/4}$ ($m$ is the index of the $m$-th zone, $\lambda$ is the wavelength and $f$ is the focal length of the lens), $M$ is the total number of zones contained in the FZP, and the modulus is used to ensure the non-negative probability. Because the standard FZPs have a pure amplitude (i.e., $\varphi_{FZP} = 0$ in $U_{FZP}$ for binary-amplitude FZPs) or phase (i.e., $A_{FZP}$ is a constant in $U_{FZP}$ for binary-phase FZPs) modulation, we only need to calculate $\Delta A$ or $\Delta\varphi$ in Eq. (1) for most objective chips. According to its definition, the deviation probability $p_1$ ranges from 0 to 1, leaving a probability $p_2 = 1 - p_1$ for the unchanged part. It behaves like an information channel with binary values, where the entropy $S = -\sum_{i=1}^{2} p_i \log_2(p_i)$ is usually used to evaluate the disorder of this information system[20].

Similarly, based on our defined probability $p_1$ and $p_2$, we can also calculate the disorder of an objective chip by using the information entropy $S$. When $p_1 = 0$ or $p_1 = 1$, the corresponding $S$ equals zero, which means high certainty without any disorder. It agrees with the real cases that both objective chips with $p_1 = 0$ and $p_1 = 1$ refer to standard FZPs, where $U_\Delta = 1$. When $p_1 = 0.5$, the entropy $S = 1$, which implies the highest disorder because half of the zones are reversed randomly. Although high disorder offers large degrees of freedom for optical super-focusing, it also destroys the imaging capability of the objective chip due to optical aberration dominated by its random $U_\Delta$. Thus, we infer that the entropy $S$ is symmetric about $p_1 = 0.5$, where the peak is located. At the side of $0 \le p_1 \le 0.5$, the entropy $S$ increases monotonically from 0 to 1. High certainty at $S = 0$ is helpful in wide-field imaging, but high disorder at $S = 1$ is required to realize super-focusing.

To obtain a good balance between imaging and super-focusing, we introduce a thermodynamic analog that the change of entropy for an isolated system originates from outside work or heat transfer[21]. In our case, a virtual work $W$ is assumed to govern the change $\Delta S$ of entropy by following a straightforward relationship $W \propto \Delta S$. The change of entropy from $S = 0$ to $S_0$ requires the virtual work $W_1$, while the change from $S_0$ to $S = 1$ requires virtual work $W_2$. We suggest that, when both virtual works are equal, i.e., $W_1 = W_2$, a good balance between imaging and super-focusing is achieved with equality $\Delta S_1 = \Delta S_2$ (i.e., $S_0 - 0 = 1 - S_0$), leading to the equilibrium point $S_0 = 0.5$. Interestingly, at this equilibrium point, its relative deviation probability $p_1$ equals 0.11, which is smaller than the middle point $p_1 = 0.25$ of the interested range $0 \le p_1 \le 0.5$. This implies that the entropy $S$ is more sensitive to the intrinsic disorder of the objective chip than the deviation probability $p_1$, thereby indicating the rationality of the proposed equilibrium point $S_0 = 0.5$.

### Strehl ratio and focal size of the objective chip

To quantitatively investigate the imaging and focusing properties of an objective chip with different disorders, a binary-phase objective chip (Fig. 1a) with a focal length $f = 1$ mm and $NA = 0.9$ (implying its diameter of 4.13 mm) is exemplified here to enhance the optical efficiency in both focusing and imaging. Compared with its corresponding binary-phase FZP, this proposed objective chip has only the phase deviation of $\Delta\varphi$ because of $\Delta A = 0$, leaving $U_\Delta = e^{i\Delta\varphi}$. This implies that any binary-phase objective chip can be taken as a combination of an analytical FZP and an additional few-ring phase (i.e., $\Delta\varphi$) mask, as sketched in Fig. 1b. Since the phase $e^{i\Delta\varphi}$ in the few-ring mask introduces optical aberration, we evaluate its influence on imaging quality by using the relative Strehl ratio ($SR$)[3]

$$SR = \frac{I_{chip}(0,0,z=f)}{I_{FZP}(0,0,z=f)} = \frac{\left| \int\int E_0(r,\varphi) \cdot e^{i(\varphi_{FZP} + \Delta\varphi)} \frac{\exp(ikR)}{R^3}(ikR-1)zrdrd\varphi \right|^2}{\left| \int\int E_0(r,\varphi) \cdot e^{i\varphi_{FZP}} \frac{\exp(ikR)}{R^3}(ikR-1)zrdrd\varphi \right|^2} \quad (2)$$

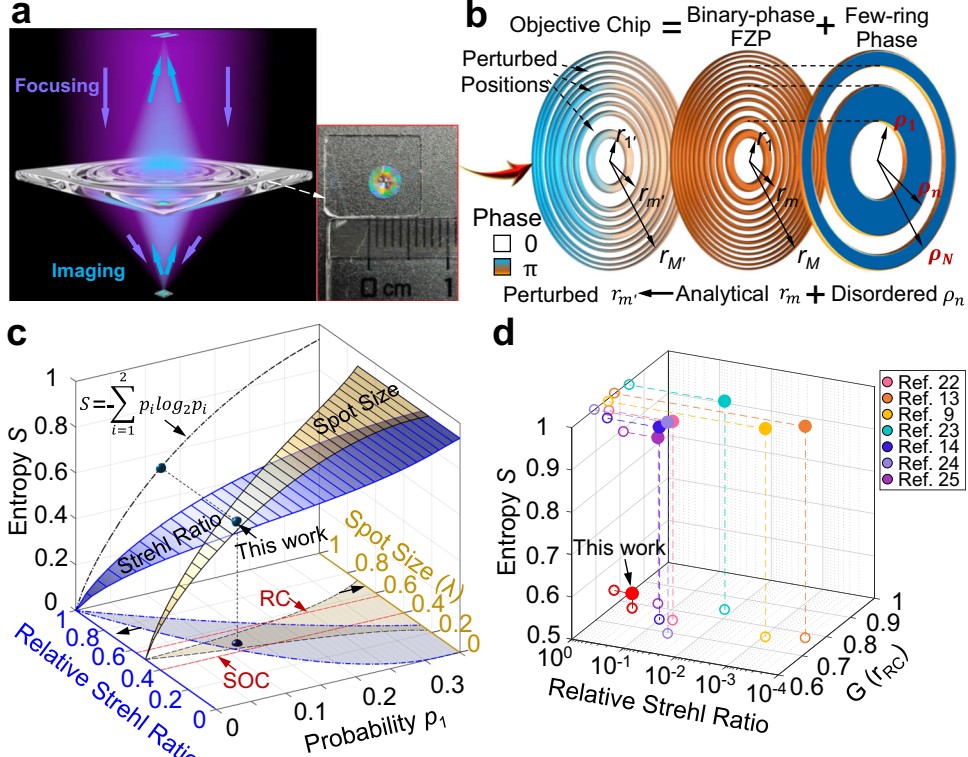

**Fig. 1 | Working principle of the bifunctional objective chip. a** Sketch of the objective chip with both focusing and imaging functionalities. **b** Design principle of the objective chip is composed of a FZP and a few-ring phase with the weak disorder. **c** Simulated relative Strehl ratio and spot size under different deviation probabilities of $0 \leq p_1 \leq 0.35$. The information entropy $S$ is also provided together as a function of $p_1$. The correlation among the Strehl ratio (blue), $p_1$ and $S$ yields a three-dimensional plotting that visualizes the underlying link between the Strehl ratio and $S$ straightforwardly. Similarly, the relationship between spot size (yellow) and $S$ is also shown with a three-dimensional configuration. Both three-dimensional drawings are projected to the longitudinal plane of $S = 0$ for a better observation,

where the Rayleigh criterion (RC) and super-oscillation criterion (SOC) are shown in red-dashed lines to distinguish the subdiffraction-limit focusing. **d** Entropy $S$, relative Strehl ratio and focal size of other reported planar diffractive lenses that provide the structural parameters (which are used to output $p_1$ for calculating entropy $S$) in their publications. For a fair comparison, all the focal sizes are normalized to the Rayleigh diffraction limit of $r_{RC} = 0.51\lambda/NA$ (in terms of FWHM). The solid circles denote these three parameters of all these reported lenses (distinguished by colors), while the hollow circles are their projections on different two-dimensional planes for a clear demonstration. Its extended version with more data is provided in Supplementary Fig. 5b for better observation.

where the incident electric field $E_0$ is taken as unity in this work, the wavenumber $k = 2\pi/\lambda$, $R^2 = (u-x)^2 + (v-y)^2 + z^2$, $r^2 = x^2 + y^2$, $\tan\varphi = y/x$, $x$ and $y$ are the Cartesian coordinates at the initial plane of the objective chip, $u = v = 0$ and $z = f$ are used to obtain the on-axis intensity at the focal plane, and $I_{FZP}$ is fixed once the FZP is given. In Eq. (2), the on-axis intensity of the FZP, having the same modulation (phase or amplitude) type as that of the designed objective chip, is used as the denominator to avoid the influence of the focusing efficiency of the FZP with different modulation types. Therefore, Eq. (2) defines a relative Strehl ratio, which is more useful in evaluating the optical aberration of imaging systems.

For a binary-phase objective chip with its deviation probability $p_1$ (only $0 \leq p_1 \leq 0.5$ is considered in the following because the entropy $S$ is symmetric about $p_1 = 0.5$), its relative Strehl ratio can be approximated as $SR = \left(1 - 2a_0 p_1 M/\sqrt{I_{FZP}}\right)^2$, where the on-axis intensity $a_0$ of each zone plate ranges from $a_{0\min} = 0.87$ to $a_{0\max} = 1$, see the detailed derivations in Supplementary Section 1. Thus, for a given $p_1$, we can analytically obtain the range of $SR$: 1) $SR_{\min} = \left(1 - 2a_{0\max}p_1 M/\sqrt{I_{FZP}}\right)^2$ for $0 \leq p_1 \leq \sqrt{I_{FZP}}/(2Ma_{0\max})$ and $SR_{\min} = 0$ for $\sqrt{I_{FZP}}/(2Ma_{0\max}) \leq p_1 \leq 0.5$; 2) $SR_{\max} = \left(1 - 2a_{0\min}p_1 M/\sqrt{I_{FZP}}\right)^2$ for $0 \leq p_1 \leq \sqrt{I_{FZP}}/(Ma_{0\max} + Ma_{0\min})$ and $SR_{\max} = \left(1 - 2a_{0\max}p_1 M/\sqrt{I_{FZP}}\right)^2$ for $\sqrt{I_{FZP}}/(Ma_{0\max} + Ma_{0\min}) \leq p_1 \leq 0.5$. By visualizing $SR_{\min}$ and $SR_{\max}$, Fig. 1c illustrates the correlation between the Strehl ratio and the information entropy $S$ by using an intermediate parameter $p_1$. With

increasing $S$, both $SR_{\min}$ and $SR_{\max}$ decrease, implying large aberration and poor imaging quality. However, its increased range ($\Delta SR = SR_{\max} - SR_{\min}$) indicates high uncertainty of $SR$, which has echoes of high disorder for a large $S$. Therefore, a small $S$ with low disorder is preferred in imaging, which needs a large $SR$.

Similarly, its focal spot is also controlled by the entropy $S$ or $p_1$. Although it is difficult to derive the spot size analytically, we obtain its upper ($r_{\max}$) and lower ($r_{\min}$) boundaries numerically (see Supplementary Section 1), as illustrated in Fig. 1c. For the larger entropy $S$, the spot size is valued in the wider range, which offers more opportunities to realize super-focusing. Hence, a large entropy $S$ with high disorder is required in super-focusing. These results reveal that imaging and super-focusing have completely opposite requirements for the entropy $S$ of an objective chip, which doubly confirms that the entropy $S_0 = 0.5$ leads to good balance.

### Design of objective chip

Since the FZP contained in the objective chip is given with analytically described structures, we only need to optimize its deviation part $U_\Delta = e^{i\Delta\varphi}$, which refers to a few-ring phase mask (see Fig. 1b). Considering that the ideal equilibrium point $S_0 = 0.5$ has a small deviation probability, only 5 rings are used to avoid introducing high disorder. In our design, all radii $\rho_n$ ($n = 1, 2, ..., 5$) of these 5 rings are chosen from the radii of zones in the FZP, which means that no new finer structure is created when combining the FZP and this 5-ring phase mask. Such a design strategy has twofold significance. First, it

greatly enhances the speed of optimization. Because all the structural details of the objective chip are given by the radii $r_m$ of the FZP, we can calculate the electric field of each zone ahead of optimization and then store it as a database for quick reading during the design, thus avoiding repeated calculation. For example, to design our objective chip with 6387 zones, it takes only 6 min to run 500 iterations by using the particle swarm algorithm in a personal computer, enhancing the speed by a factor of ~$3.8 \times 10^5$ (see the calculation details in the Methods). Second, it allows large-scale and low-cost fabrication of our objective chip because its smallest feature size is theoretically fixed to $\Delta r_{\min} = \lim_{m \to \infty} (r_m - r_{m-1}) = \lambda/2$.

The 5-ring phase mask in our objective chip is designed with $\rho_1 = r_{283}$, $\rho_2 = r_{850}$, $\rho_3 = r_{1046}$, $\rho_4 = r_{1258}$ and $\rho_5 = r_{6391}$, which yields $p_1 = (567 + 212)/6391 = 0.122$ and entropy $S = 0.535$. Compared with other reported PDLs[9,13-15,22-24], the achieved entropy in our objective chip is the closest to the equilibrium point $S_0 = 0.5$ (see Fig. 1d), implying its advantages in balancing optical imaging and superfocusing. It exhibits the fundamental difference of our objective chip from other lenses with their entropies approaching 1. Furthermore, the relative $SR = 0.45$ of our objective chip is also the highest among those lenses, which theoretically predicts the best imaging capability. The optimized focal spot has a lateral FWHM (i.e., full width at half maximum) of 180 nm ($0.44\lambda$, below the Rayleigh criterion of $r_{RC} = 0.51\lambda/NA = 0.57\lambda$), which is slightly larger than the superoscillation criterion $r_{SOC} = 0.358\lambda/NA = 0.398\lambda$ (in terms of FWHM)[25] to avoid strong sidebands in a superoscillatory spot. The simulated electric fields near the focal plane are provided with experimental results for a better comparison, as shown later.

## Fabrication of objective chip

The possibility of employing semiconductor processes to fabricate planar flat optics will ultimately allow mass production of flat optics at a low cost and push this technique for wide market adoption. As discussed above, our design strategy makes low-cost and fast DUV lithography feasible for fabricating a 4 mm-diameter objective chip. Under the conditions of $\lambda = 405$ nm, $f = 1$ mm and $NA = 0.9$ in this work, $\Delta r_{\min}$ is only 225 nm, which is within the capability of commercial DUV lithography (with a critical dimension of 200 nm). This key feature will greatly facilitate the future manufacturing of our objective chips by reducing the cost compared with other flat lenses with critical dimensions smaller than 200 nm that would require much more costly 12-inch immersion lithography[26]. We note that E-beam lithography is able to write the pattern at high resolution; however, it is not the technique for large-scale fabrication due to its speed limit and the inevitable stitching error (with a small writing square of several tens of micrometers). The fabrication details of our objective chip are provided in the Methods.

The inset of Fig. 1a shows the microscopic image of our fabricated objective chip, where different colors arise from optical scattering of daylight. To reveal the fine details, we show scanning-electron-microscopy (SEM) images of the objective chip in Fig. 2a, where both simulated and experimental widths from the center to the outermost boundary are also provided with a maximum deviation of < 75 nm. The possible reason comes from insufficient exposure time of photoresist under DUV radiation, which can be solved by increasing the exposure time. Using a profilometer, we characterize the groove depth of ~530 nm (Fig. 2b) that yields a phase modulation of ~$1.23\pi$, which is larger than the ideal value of $\pi$ due to overetching issues. Nevertheless, we emphasize that such a deviation of $0.23\pi$ in phase modulation leads to a theoretical decrement of only ~2.6% in the focusing efficiency of the objective chip, see the simulation details in the Methods.

## Subdiffraction-limit focusing by objective chip

To verify the focusing capability of the objective chip, we first measure the diffraction field near the focus of the objective chip

under the illumination of collimated circularly polarized light by using a 0.95-$NA$ objective lens, see the experimental details in Supplementary Section 3. Figure 2c shows the $x$-$z$ and $y$-$z$ cross-section of the measured light intensity, which reveals a record-long focal length of 1 mm (compared with the previous planar diffractive lenses[9,14,15,22,27,28]). The axial depth of focus (DOF) is extended from the simulated ~500 nm to the experimental ~5500 nm, which is caused by the fabrication error of the groove width in the objective chip (see Fig. 2a). Such a long DOF offers good tolerance to sample alignment in an SCM[13]. Fortunately, the focusing spot size of the objective chip is less influenced by the weak phase perturbation from the imperfect zones, as confirmed by the good agreement between the simulated and experimental line-scanning intensity profiles (Fig. 2d). To quantitatively evaluate the focusing effect, we present the experimental FWHM of the focal spot size along the propagation of light in Fig. 2e, indicating the varying FWHM from 170 nm to 210 nm (tightly close to the simulated 180 nm). Compared with the Rayleigh criterion of $0.51\lambda/NA$, these achieved focal spots confirm the subdiffraction-limit focusing capability of the proposed objective chip. Due to its supercritical feature with lateral FWHMs above the superoscillation criterion[25], the focusing spots have no strong sideband, as observed in Supplementary Movie 1, which dynamically records the focusing process near the focal plane.

The focusing efficiency, defined as the ratio of the focused power (experimentally filtered by a 150 µm-diameter pinhole at the focal plane) to the total power incident on the objective chip, is measured to be 12.3% (see the measurement details in Supplementary Section 4), which is lower than its theoretical efficiency of 18.7% (see its calculation in the Methods), as shown in Fig. 2f. To investigate its influence from etching depth, we also fabricate two additional objective chips with etching depth of 350 nm and 390 nm (with the phase delay of $0.81\pi$ and $0.905\pi$ respectively), where the measured efficiency of 13.8% and 15.4% (slightly lower than the theoretical values, see Fig. 2f). This discrepancy in efficiency is attributed to incomplete constructive interference of light diffracted from two neighboring zones because of the insufficient etching widths. Although our achieved efficiency is not as high as those of traditional objectives and metalenses, it still exhibits significant enhancement in comparison with those of amplitude-type zone-plate lenses[9,14,15,22,28].

## Direct wide-field imaging by objective chip

To investigate its wide-field imaging, a knife-edge object (see its microscopic image at the left-bottom of Fig. 2g) is located at a distance of $z = 1.2f$ from the objective chip[29]. After illuminating the knife-edge object, the transmitted light is collected by our objective chip. According to the imaging formula of a lens, we can roughly estimate its imaging distance of $6f$ at the other side of the objective chip, thereby exhibiting an imaging magnification of 5×. The recorded image of the knife-edge object is shown at the bottom-right panel of Fig. 2g, which reveals a well-defined boundary at the edge. A dynamic Movie that records its imaging process by tuning the axial position of such a knife-edge object is provided in Supplementary Movie 2, which is captured in a homemade measurement setup (see Supplementary Fig. 8 and Supplementary Section 5).

The modulation transfer function (MTF) of this objective chip is characterized by using the line-scanning intensity across the edge in the image. To decrease the experimental error, we employ a mean of the line-scanning intensity (with its spatial dimension scaled down by a factor of its magnification $M = 5$) at the red-rectangle region in the image of Fig. 2g to recover the line spread function (LSF) of this imaging configuration. After fitting the line-scanning intensity with an error function (taken as a convolution between a Gaussian function and a jump function), we carry out the deconvolution of the fitted error function, yielding the retrieved LSF (Fig. 2h). Using a

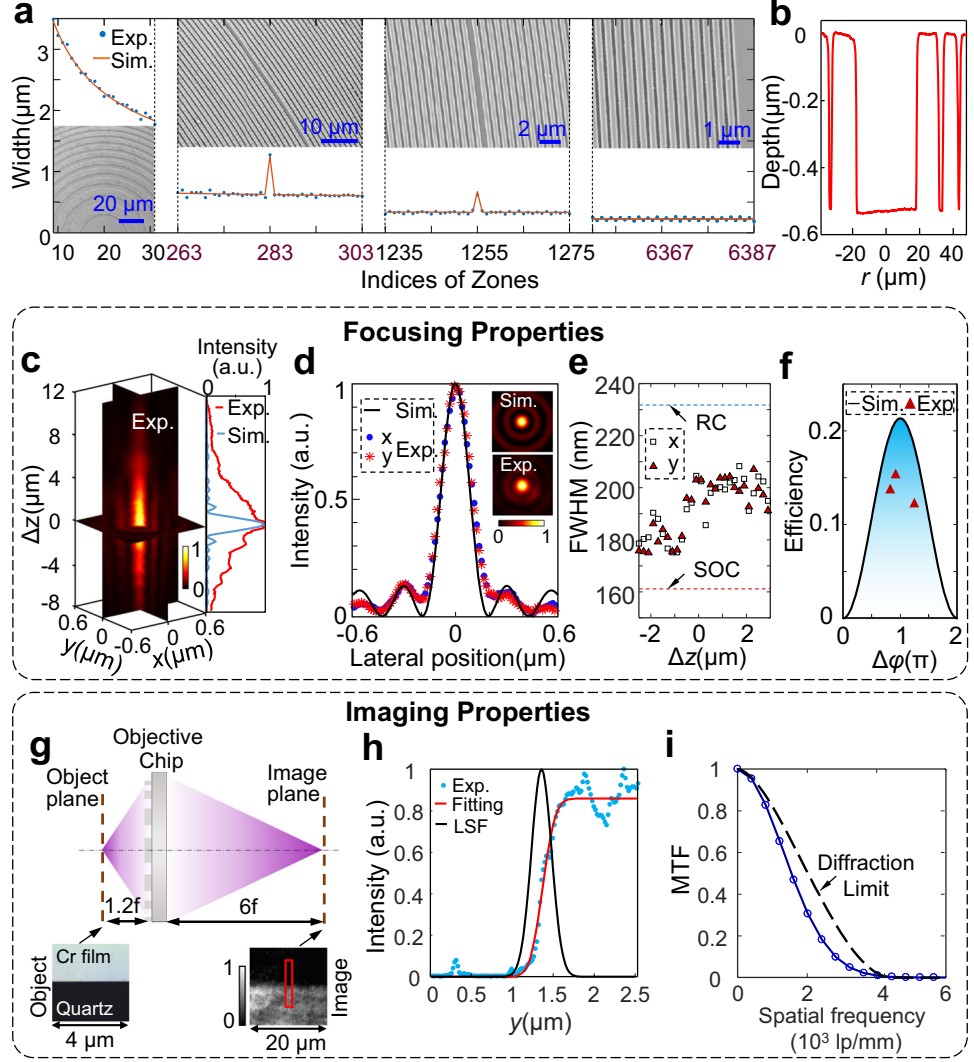

**Fig. 2 | Focusing and imaging properties of our objective chip. a** Simulated and experimental widths of belts at the different regions of our objective chip. Inset: SEM images of the different regions by addressing the corresponding zone numbers. **b** The etched depth around the center region of the objective chip is measured experimentally by using a profilometer. **c** Optical field near the focal plane of the objective chip. Cross sections of the measured intensity profiles are shown in the left panel, while the right panel shows a comparison between the simulated and experimental on-axis line-scanning intensity profiles. **d** Simulated and experimental line-scanning intensity profiles at the focal plane of the objective chip. Their 2-dimensional intensity profiles in the region of 1.2 μm × 1.2 μm are provided in the insert. **e** Lateral FWHMs of the measured spot near the focal plane. RC: Rayleigh criterion ($0.51\lambda/NA$); SOC: Superoscillation criterion ($0.358\lambda/NA$). **f** Simulated and experimental efficiency when the phase difference $\Delta\varphi$ between two neighboring etched and unetched parts changes from 0 to 2 π. **g** Sketch for wide-field imaging by using our objective chip. The object and image distances are 1.2 $f$ and 6 $f$ (the focal length $f$ = 1mm) respectively, yielding a magnification of 5×. Such a magnification is chosen to avoid optical aberration, while it is enough to demonstrate the capability of collecting light with high spatial frequencies. Inserts: the knife-edge object (left, captured by using a reflective microscope that generates bright chromium film and dark quartz substrate) and its image (right) taken by using our objective chip. **h** Experimental edge spread function (ESF, which is calculated by using the average intensity along the long side of the red box in the insert of (**g**)). To evaluate its resolving power, the spatial coordinate $y$ is scaled down by its magnification of 5. The experimental ESF is fitted by an error function, the deviation of which outputs the line spread function (LSF). **i** Retrieved modulation transfer function (MTF, solid-circle curve) of the objective chip by using the Fourier transform of the achieved LSF in (**h**). The diffraction limit is also provided for a better comparison.

Fourier transformation of the retrieved LSF, we finally obtain its MTF (Fig. 2i), which indicates a cut-off frequency of 4000 lp/mm. This implies an imaging resolution of 250 nm, which corresponds to an effective $NA$ of 0.83 (evaluated by using $0.51\lambda/NA_{eff}$ = 250nm) for imaging. Compared with the previous metalens with a cut-off spatial frequency of 2000 lp/mm[4], our objective chip achieves a twofold enhancement in resolving power when operating in the mode of direct wide-field imaging. These experimental results have confirmed that our objective chip has sufficient imaging ability to collect high spatial frequencies from fine details of objects, which is superior to all previous superoscillation[9,10,12,22,27,28] and supercritical lenses[13–15].

## Objective-chip-based reflective SCM

A high-resolution reflective SCM (Fig. 3a, see its working principle in the Methods) has been built successfully due to both features (i.e., subdiffraction-limit focusing and direct wide-field imaging) of our objective chip. First, the enhanced focusing efficiency allows more light to illuminate the object in a reflective mode, which underpins subsequent collection and detection of reflected light. Figure 3b shows the experimental signals detected by the photomultiplier tube (PMT) and charge-coupled device (CCD) when the nano-objects are moved longitudinally near the focal plane of $z$ = 1mm. It reveals that the PMT signal reaches its maximum for the in-focus (i.e., $\Delta z$ = 0) nano-objects and decreases gradually with the increment of the out-of-focus

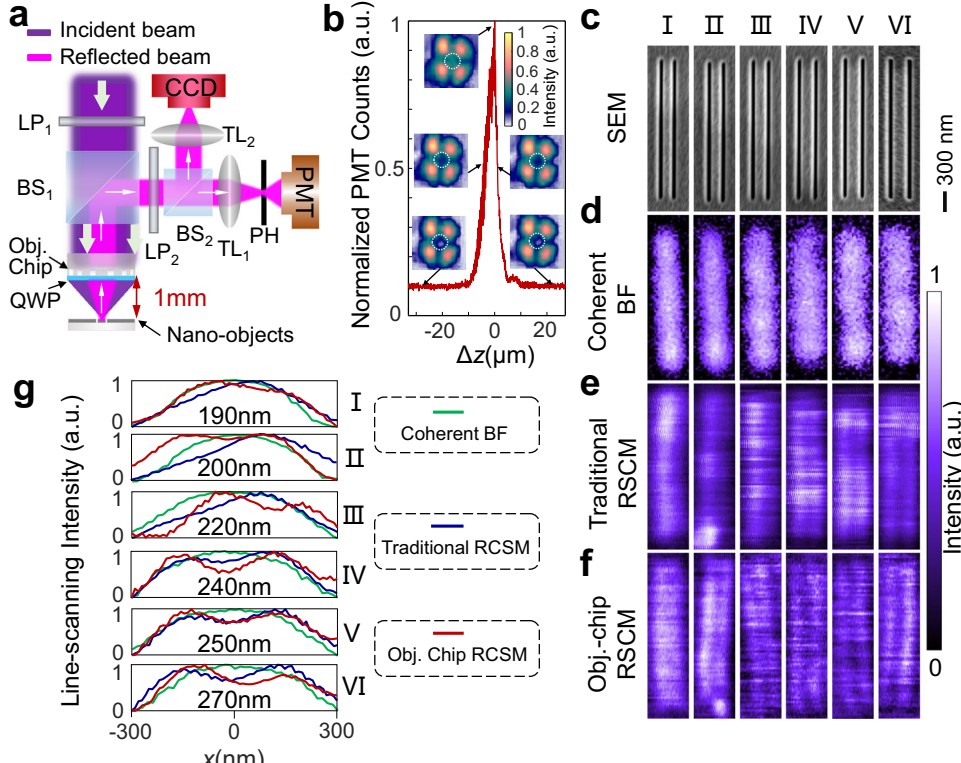

**Fig. 3 | Reflective scanning confocal microscopy based on our objective chip.**
**a** Sketch of the optical setup of the objective-chip-based reflective SCM. LP: linear polarizer; BS: beam splitter; QWP: quarter wave plate; TL: tubelens; PH: pinhole; PMT: photomultiplier tube. **b** Detected signals (PMT, curve) and images (CCD, inserts) when the nano-objects are scanned with the out-of-focus distance $\Delta z = z - f$. In the CCD images, the dashed circles denote the position of the focused signal light. In the PMT signals, the nonzero background (-0.1) is caused by the incompletely suppressed light (i.e., the four-lobe patterns) reflected from the back surface of the objective chip. **c**–**f** Double slits (**c**, SEM) and their images by using

coherent bright-field microscopy (**d**), traditional RSCM (**e**) and objective-chip-based RSCM (**f**). The CTC distances of double slits I to VI are 190 nm, 200 nm, 220 nm, 240 nm, 250 nm and 270 nm, respectively. The height and width of each slit are 2 μm and 50 nm, respectively. Scale bar: 300 nm. The working distance and magnification of the objective used in coherent BF microscope and traditional RSCM is 310 μm and 100 ×, respectively. For a better comparison, the pictures in (**b**) are scaled down by its magnification of 255.3 × . **g** Line-scanning intensity profiles of images by using different microscopies.

distance $|\Delta z|$, which is doubly checked by the CCD images (see the inserts in Fig. 3b and Supplementary Movie 3). The FWHM of the PMT measured intensity profile is ~5 μm, which agrees with the experimental DOF of 5.5 μm (see Fig. 2b, d). This result confirms that our objective chip can efficiently focus the incident beam and collect the reflected light. Second, a powerful imaging ability with an effective $NA$ of 0.83 is required to enhance the practical imaging resolution of an SCM. Theoretically, we have already shown that the resolution of an SCM is less influenced by the $NA$ of the collection objective[7], which, however, is valid only for infinitesimal point objects. For real objects with finite sizes ranging from tens to hundreds of nanometers, the $NA$ of the collection objective should be larger than 0.7 for a better resolution in an SCM, see the simulated proofs in Supplementary Section 6. Due to these two features of our objective lens mentioned above, the subdiffraction-limit focusing and the high-resolution imaging enable a reflective SCM. More experimental details about the scanning imaging are provided in the Methods.

To test its resolution, we provide the imaging results of 50 nm-width and 2 μm-length double slits with center-to-center (CTC) distances ranging from 190 nm to 270 nm. As shown in Fig. 3c, these slits are etched on a 140 nm-thick chromium film on a quartz substrate. Using a 0.9 NA objective for a fair comparison, the coherent bright-field microscope cannot resolve these double slits (Fig. 3d) while conventional reflective SCM can only resolve the double slits with CTC distances larger than 240 nm (Fig. 3e). In contrast, our objective-chip-based reflective SCM has an enhanced resolution so that double slits with a CTC distance of 200 nm can be distinguished with a valley of

intensity in the image (Fig. 3f), where the distortion is caused by mechanical variation of the sample. The qualitative comparisons among their line-scanning intensity profiles (Fig. 3g) doubly verify an imaging resolution of 200 nm achieved by our SCM. In addition, all these experimental results regarding scanning images are confirmed by our simulations (see Supplementary Section 7 and Supplementary Fig. 10) with the theory of SCM[7,30,31].

Complex nano-objects can also be imaged with a high resolution by using our SCM. Figure 4a shows the SEM image of a dolphin (composed of 50 nm-width curves) with a total size of 8 μm × 8 μm. Due to their limited resolutions, both coherent BF microscopy and traditional SCM can map only rough contours of the dolphin but lose fine details, such as the eye and tail (Fig. 4b, c). In comparison, our SCM can clearly resolve all these fine details (Fig. 4d) with a narrower line width (Fig. 4e). Furthermore, two lines with a CTC distance of 225 nm (see the dashed-red lines at the lower rows in Fig. 4b–d) in the tail can be distinguished only by using our SCM. The low contrast of intensity in the image for our SCM comes from the relatively lower focusing efficiency of the objective chip in comparison with the traditional objective. However, it has little influence on the resolution and clarity of the image, as observed in Fig. 4d.

## Discussion
Among all the PDL-based SCMs, our current SCM has the advantages of eliminating bulky objectives, a millimeter-level working distance, reflection-mode operation, working for both transparent and non-transparent substrates, and a competitive resolution of 0.49 λ, as shown

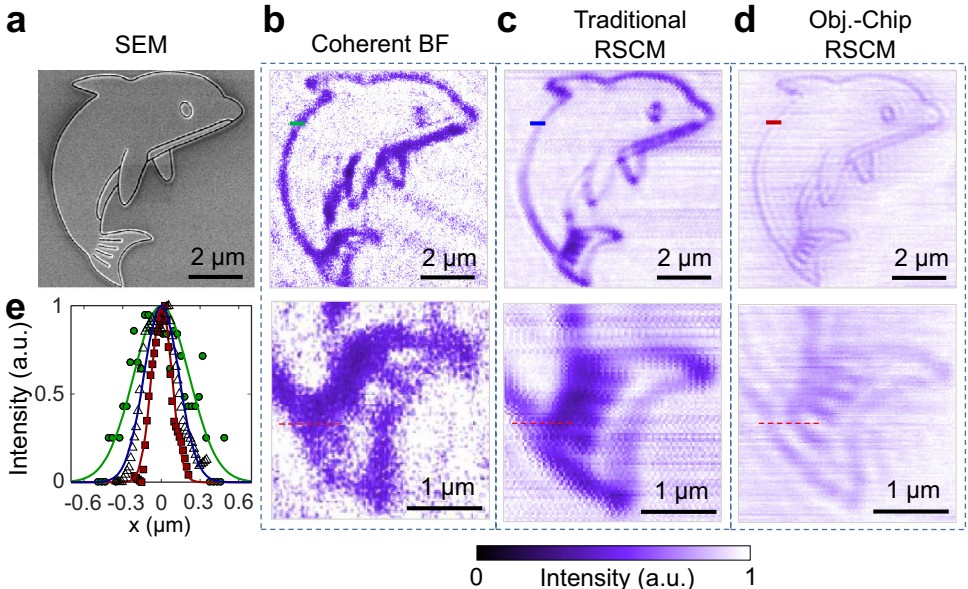

**Fig. 4 | Imaging complex nano-objects by using different microscopies. a** SEM image of a "dolphin" object composed of 50 nm-width curves. **b–d** Images (upper row) by using coherent bright-field microscopy (**b**), traditional RSCM (**c**) and objective-chip-based RSCM (**d**). Their zoomed-in images with more details are shown in the lower row. In (**b**, **c**), all the parameters (i.e., the working distance and magnification of objectives used in coherent BF microscope and traditional RSCM) are the same as those in Fig. 3d, e. **e** The line-scanning profiles of the imaged dolphins (along the colored lines of the upper row in (**b–d**) by using coherent bright-field microscopy (green circles), traditional RSCM (triangles) and objective-chip-based RSCM (red squares). These line-scanning data are fitted by using a Gaussian shape to guide the eyes.

in Supplementary Table 1. For commercial objectives, pursuing a high *NA* and long working distance simultaneously leads to an increment in the diameters of optical elements and the accompanying optical aberrations that need large-scale nonspherical surfaces for correction[3], thereby yielding extremely high costs in both the fabrication and design of elements. This issue does not exist in our objective chip, where the smallest feature of $\lambda/2$ will not change with increasing *NA* and focal length, so the same fabrication tools and design methods reported in this work can also be used to develop more advanced objective chips with even larger *NA*s and longer focal lengths.

Our objective chip has a fabrication cost of ~$42 dollars (estimated by the total price of 300 chips in an 8 inch quartz wafer, see Supplementary Fig. 11), which is ~100 times cheaper than the price of commercial objectives. This objective chip has a volume of 4 mm × 4 mm × 0.5 mm, which indicates a shrinking factor of 4300 (~3.6 orders of magnitude) compared with traditional objectives (ZEISS, EC Epiplan-Neufluar 100×, *NA* 0.9, M27).

Developing this reflective configuration makes a truly important step to push the technology of PDL-based SCMs towards practical applications, because many samples have opaque substrates that are incompatible with all the previous PDL-based SCMs. Note that the objects used in this work are made in a high-reflectivity metal film, which helps to enhance the imaging contrast. If the difference between the optical reflection of the object and its surrounding background is not obvious, one should increase the optical efficiency of the objective chip and the sensitivity of the optical detector. The focusing efficiency of the objective chip can be enhanced further if multilevel phase elements[32] are used. The detector can also be updated to the single-photon level for a better recording of collected photons by our objective chip, enabling the characterization of less-reflective biological tissues and cells even in a living body.

In summary, we have proposed information entropy to evaluate the disorder of an optimized zone-plate lens. The suggested equilibrium point $S_0 = 0.5$ is used to guide the quick design of a 1 mm focal-length, high-*NA* and low-cost objective chip with efficiency-enhanced subdiffraction-limit focusing and direct wide-field imaging. These advantages open the way to demonstrate compact and high-resolution reflective SCM with PDL, which will greatly benefit from optical[33] to biomedical imaging[34,35].

## Methods

### Design and optimization details

To avoid the creation of additional finer structures when combining the FZP and the few-ring mask, each radius $\rho_n (n = 0, 1, 2, \ldots, N)$ in the *N*-ring mask is valued within the radii (i.e., $r_m$) of belts in the FZP. To highlight these selected radii in the zone plates, we label them $r_{M_n} = \rho_n$, which means that the $n^{th}$ ring of the few-ring mask has the same radius as the $M_n^{th}$ belt of the zone plate. Although it decreases the degree of freedom to design the few-ring mask, significant benefits are achieved in simplifying the optimization and fabricating the sample.

Benefiting from this design strategy, we can express the electric field of light focused by the objective chip as

$$E_{chip}(u, v, z) = \sum_{n=0}^{N-1} (-1)^n \left[ \sum_{m=M_n}^{m=M_{n+1}} (-1)^m A_m \right] = \sum_{n=0}^{N-1} \sum_{m=M_n}^{m=M_{n+1}} (-1)^{m+n} A_m$$
$$= \sum_{n=0}^{N-1} \sum_{m=M_n}^{m=M_{n+1}} (-1)^{m+n} \int_{r_m}^{r_{m+1}} \int_0^{2\pi} E_0(r, \varphi) \frac{\exp(ikR)}{R^2} \left( ik - \frac{1}{R} \right) z r dr d\varphi$$

(3)

where $A_m$ is the electric field of light diffracting from the $m^{th}$ belt in the FZP, the wavenumber $k = 2\pi/\lambda$, $R^2 = (x - u)^2 + (y - v)^2 + z^2$, $r^2 = x^2 + y^2$, $\tan \varphi = y/x$, $x$ and $y$ are the spatial coordinates at the plane of the objective chip, $x$, $y$ and $z$ stand for the spatial position of interest. Considering the nonparaxial propagation of light in such a high-*NA* objective chip, we calculate $A_m$ by using the rigorous Rayleigh-Sommerfeld diffraction integral without any approximation[7,36]. Although thousands of belts are included in this objective chip, only $N-1$ variables (i.e., $M_1$, $M_2$, …, $M_{N-1}$ because $M_0 = 0$ and $M_N = 6391$) are unknown in Eq. (3) because all $r_m$ are given. Since the phase jump of π occurs at the $M_n^{th}$ belt ($n = 1, 2, \ldots, N - 1$), both the $M_n^{th}$ and $(M_n + 1)^{th}$ belts are combined into one, leading to the belt number of $M-(N-1)$ in the final objective chip.

Since no new belt appears in this strategy, the electric field $A_m$ can be calculated ahead of optimization and then stored in a database, thus enhancing the design speed. The particle swarm algorithm[37] is used to optimize the $N$-1 parameters, see the details in Supplementary Section 2. In our design, $N = 5$ is employed with 4 unknown parameters, which can be determined with the values of $M_1 = 283$, $M_2 = 850$, $M_3 = 1046$ and $M_4 = 1258$ by carrying out 500 iterations in ~6 min in a personal computer (Intel Core i5-7500 CPU 3.40 GHz, 32 G RAM). In each iteration, the optimization algorithm contains 20 populations, each of which stands for one design of objective chip. If our design strategy with the pre-calculated database is not used, we can roughly estimate its time cost of $3.8 \times 10^4$ ($= 3.8 \times 20 \times 500$) hours to finish the design by running 500 iterations, because it will take ~3.8 h to calculate the focal field of a single objective chip by numerically integrating all the zones with Rayleigh-Sommerfeld diffraction under the same computation environment. Thus, our design strategy accelerates the optimization by a factor of $3.8 \times 10^5$. In our designed objective chip, the phase-reversed zones contain two parts: 1) from $m = 284$ to $m = 850$ and 2) from $m = 1047$ to $m = 1258$, resulting in $p_1 = (567 + 213)/6391 = 0.122$ and $p_2 = 1 - p_1 = 0.878$. According to the definition of information entropy, we have $S = 0.535$, which is tightly close to the equilibrium point $S_0 = 0.5$.

## Fabrication details

The designed objective chip is fabricated through a deep-ultraviolet (DUV, Nikon S204) exposure process. The quartz substrate is first deposited with 200 nm-thick aluminum using a physical vapour deposition (PVD) system (AMAT Endura). Then, a 300 nm-thick positive photoresist (UV135) is coated and baked. Subsequently, the photoresist is patterned using DUV lithography. After development, the aluminum film without photoresist is etched sufficiently by an inductively coupled plasma (ICP) etching system (LAM 9600). Thus, the patterns of the objective chip are transferred into the aluminum film after removing the residual photoresist. Next, using the aluminum film as a masking layer, the quartz substrate is etched for a designed thickness by an inductively coupled plasma-reactive ion etching (ICP-RIE) system (Oxford, Plasma Pro System100 ICP380). Finally, the aluminum hard mask is removed by Tetramethylammonium Hydroxide (TMAH, 2.5%) solvent, yielding the expected phase-type objective chip.

## Theoretical efficiency of the objective chip

To obtain the theoretical efficiency of the objective chip, we first calculate the electric field $A_m$ of the $m^{\text{th}}$ belt in the zone plate at the focal plane (ignoring the influence of the width error of etched belts). For the sake of convenient simulation, we calculate the one-dimensional field along the radial direction within the range of $\lambda$ (starting from $r = 0$), where the focused light is concentrated. To evaluate the experimental error, we update Eq. (3) by considering an actual phase difference of $\Delta\varphi$ as

$$E_{\text{chip}}(r) = \sum_{n=0}^{n=4} \sum_{m=M_n}^{m=M_{n+1}} e^{i \cdot \Delta\varphi \cdot \frac{1+(-1)^{m+n}}{2}} A_m \qquad (4)$$

where $M_0 = 0$, $M_1 = 283$, $M_2 = 850$, $M_3 = 1046$, $M_4 = 1258$, and $M_5 = 6391$, $\Delta\varphi = \frac{2\pi}{\lambda}(n-1)d_{\text{etch}}$ is the phase difference between etched and unetched zones in the objective chip at a wavelength of $\lambda = 405nm$, the refraction index of the quartz substrate is $n = 1.47$ and the experimental etching depth $d_{\text{etch}} = 530nm$ (referring to $\Delta\varphi = 1.23\pi$). To further evaluate the energy flux in the circular area with a radius of $\lambda$ at the focal plane of the objective chip, we employ the expression:

$$W_{\text{chip}} = \int_0^\lambda \int_0^{2\pi} I_{\text{chip}}(r) r dr d\varphi, \qquad (5)$$

where the intensity $I_{\text{chip}}(r) = |E_{\text{chip}}(r)|^2$. Similarly, we can acquire the energy flux of the standard binary-phase zone plate in the same area with $W_{\text{FZP}} = \int_0^\lambda \int_0^{2\pi} I_{\text{FZP}}(r) r dr d\varphi$, where $I_{\text{FZP}} = |E_{\text{FZP}}(r)|^2 = \left|\sum_{m=0}^{m=6391}(-1)^m A_m\right|^2$. Finally, the theoretical focusing efficiency of the objective chip can be evaluated as

$$\eta_{\text{chip}} = \eta_{\text{FZP}} \cdot \frac{W_{\text{chip}}}{W_{\text{FZP}}}, \qquad (6)$$

where $\eta_{\text{FZP}} = \text{sinc}^2(1/L)$ denotes the optical efficiency of multilevel phase elements, and $L$ is the number of phase levels. For the binary-phase FZP, we have $\eta_{\text{FZP}} = 40.5\%$. Note that, Eq. (6) depends on the geometric parameters $M_n$ and $\Delta\varphi$, which allows us to conveniently investigate optical efficiency of the objective chip. For example, once the geometric parameters are fixed in this work, we can simulate the theoretical focusing efficiency for different $\Delta\varphi$ values (see Fig. 2e), which reveals a peak efficiency of 21.3% at $\Delta\varphi = \pi$. For the experimental $\Delta\varphi = 1.23\pi$, its corresponding theoretical efficiency is 18.7% with a deviation of 2.6% from the peak efficiency. In fact, due to the insufficient etching width, the focusing efficiency will further decrease, which is the reason why measured efficiency is lower than theoretical efficiency (Fig. 2f).

## Work principle of objective-chip-based reflective SCM

A schematic diagram of the objective-chip-based reflective SCM is given in Fig. 3a. The confocal configuration consists of our objective chip and two tube lenses (TL$_1$ and TL$_2$), where their focal planes are conjugated with that of the objective chip. The objective chip is illuminated by a collimated light beam with a wavelength $\lambda = 405nm$.

To increase the signal-to-noise ratio of the entire system, we suppress the light reflected from the back-surface (i.e., the bare-quartz side without any structure) of the objective chip by utilizing two orthogonal linear polarizers (LP$_1$ and LP$_2$) and a quarter-waveplate (QWP) thin film, as sketched in Fig. 3a. Because both LP$_1$ and LP$_2$ have orthogonal transmission directions, the reflected light from the back-surface of the objective chip is blocked efficiently, leaving a very weak background with a four-lobe pattern (see the insert in Fig. 3b). It is important to note that such a four-lobe pattern has a dark center, where the signal light reflected by the objects is located, thus leading to spatial separation between the noise and signal. The 60 μm-thick QWP thin film (with an angle of 45 degrees between its fast axis and the transmission directions of both LPs) is adhered to the structure side of the objective chip to obtain circular polarization (CP), which enables us to realize a circularly symmetric focal spot for isotropic scanning of the image. Moreover, the reflected CP signal light from nano-objects passes through the QWP thin film again and is converted into linear polarization with its direction aligned to the transmission direction of LP$_2$. Therefore, the second linear polarizer (LP$_2$) can block the noise light reflected from the back surface of the objective chip and transmit the signal light efficiently, thus increasing the signal-to-noise ratio of the PMT signals.

After tuning the signal, we coarsely move the objective chip mounted on the electric stage toward the nano-objects. When the nano-objects are close to the focal plane, the PMT signals behave like that shown in Fig. 3b, having a peak when the nano-objects are in focus. At the same time, the recorded pattern at the CCD becomes the smallest. Due to the conjugation relationship between the objects and the PMT, we can adjust the PMT to collect an optical signal at the focal plane of TL$_2$ filtered by a 10 μm-diameter pinhole. Then, we finely move the nano-objects mounted on the 3D piezo stage to the focal plane by observing the signal collected by the PMT. When the PMT signal reaches its maximum (see Fig. 3b), we assume that the nano-objects are at the focal plane of the objective chip. Finally, the nano-objects can be

scanned at the focal plane, and the signal collected by the PMT can be recorded simultaneously to complete the scanning image.

### Experimental details about the scanning imaging

The 3D piezo stage (PI-545.3R8S) and the controller (PI-E727) are integrated into a single device with a scanning resolution of ~1 nm. The PMT has the module of the WiTec 3000 R series. We utilize the Lab-VIEW language to control the movement of the 3D piezo stage and read the signal collected by the PMT with a DAQ card (NI USB-6000, 12-bit, sampling rate: 10 Ks/s) simultaneously. A 10 μm-diameter fiber (Thorlabs M64L01 10 μm 0.1 NA) is employed here as the pinhole.

In the experiment testing the imaging resolution, the scanning range of double slits is $3 \mu m \times 1 \mu m$ with $100 \times 100$ pixels, which takes ~15 min to finish one image. The scanning speed can be updated further by using a high-speed stage and digital-analog converter. The $2.4 \mu m \times 0.6 \mu m$ range of scanning images in Fig. 3c−f is employed to fully cover the objects. In the experiment for imaging complex nano-objects, the scanning range is $9 \mu m \times 9 \mu m$ with $150 \times 150$ sampling points, which takes ~34 min. Only the $8 \mu m \times 8 \mu m$ range is shown in Fig. 4b−d to highlight more details of the images.

### Data availability

The all data generated in this study have been deposited in the Zenodo database (https://doi.org/10.5281/zenodo.8268113)[38].

### Code availability

The codes for designing objective chip and simulating focus profile and imaging have been deposited in the Zenodo database (https://doi.org/10.5281/zenodo.8268113)[38].

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

### Acknowledgements

K.H. thanks the CAS Project for Young Scientists in Basic Research (Grant No.YSBR-049), the National Natural Science Foundation of China (Grant No. 12134013), the National Key Research and Development Program of China (No. 2022YFB3607300), the CAS Pioneer Hundred Talents

Program, and support from the University of Science and Technology of China's Centre for Micro and Nanoscale Research and Fabrication. J.T. thanks the A*STAR AME IRG program (Grant No. A2083c0058).

## Author contributions

K.H., J.T., and C.Q. conceived the idea. J.H. and K.H. performed the simulations. D.Z. and H.L. prepared and fabricated optical samples. J.H. and D.Z. built up the experimental setup and performed the characterization. J.H., K.H., C.Q., and J.T. wrote the manuscript. K.H., J.T., and C.Q. supervised the overall project. All authors discussed the results, carried out the data analysis and commented on the manuscript.

## Competing interests

The authors H.L., J.T., and C.Q. declare no competing interests. The authors J.H., K.H., and D.Z. declare the following competing interests. J.H., K.H., and D.Z. have filed two patent applications related to this work through University of Science and Technology of China. The first patent (Jun He, Kun Huang, Dong Zhao, "A planar objective and its fabrication method", 202211555521.4(2022), under review) has been applied by University of Science and Technology of China. It refers to the design and fabrication of the objective chip for the simultaneous realization of super-focusing and direct wide-field imaging. The second patent (J.H., K.H., D.Z., "A reflective scanning confocal microscopy based on planar objective", 202211563120.3(2022), under review) has been applied by University of Science and Technology of China. It refers to a reflective scanning confocal microscopy based on the objective chip for sub-diffraction-limit imaging.
