## [Peer Review File · Nature Communications]

An entropy-controlled objective chip for reflective confocal microscopy with subdiffraction-limit resolutionReviewer #1 (Remarks to the Author):

The authors have demonstrated a planar diffractive lens that has both super-focusing and wide-field capabilities, which are challenging to be realized in a single planar device. To solve the problem of realizing super-focusing and wide-field issues, they introduce the concept of information entropy to evaluate the disorder of such optimized structures, which generally destroy the wide-field imaging. By using the theory of entropy change, they have suggested a special point $S=0.5$, which is considered as a good point of balancing the super-focusing and wide-field imaging. Thus, they propose a theoretical criterion to realize the bifunctional capabilities. By using the detailed optimization, they design and fabricate the expected lens with the entropy $S=0.53$, which is close to the ideal point $S=0.5$. They have also verified the super-focusing and wide-field imaging in experiment, which has good agreement with the simulations. It verifies the feasibility of their proposed method. Then, the authors have also demonstrated the planar-lens-based scanning confocal microscopy with sub-diffraction-limit resolution, which removes the usage of traditional objective lens and therefore is superior to all the reported planar lenses. So, the novelty of this manuscript is high and the experimental and theoretical results are solid for publications in Nature Communications. I might recommend its publications in Nat. Commun. if the following issues are clarified.

1. The proposed objective chip is based on the binary-phase FZP. If the objective chip is based on the pure-phase metalenses, how can one implement the design and optimization of such bifunctional objective chips?
2. In the design of objective chips, the authors have used the 5-ring phase mask. Is it possible to discuss whether it is possible by using the phase mask with different (4 or even smaller, 6 or even larger) rings?
3. The fabricated objective chip has the deviation of 75 nm, which is small compared with the total size of the chip. But, such a deviation is large enough if it is compared with the local width of the outermost rings. So, it is better to provide more data or discussions about such fabrication errors.
4. The authors have suggested that the focusing efficiency of objective chip can be even enhanced by using high-efficiency dielectric metasurfaces. Indeed, it could increase the focusing efficiency, but the imaging efficiency decreases because the large-angle rays from the objects have small conversion efficiency (as pointed out in the introduction part). So, it is better to suggest the efficient method to enhance the efficiency of both focusing and imaging. Or removing the suggestion of "high-efficiency dielectric metasurfaces".
5. The scanning speed is slow now (100*100 pixels takes 15 minutes). Although the authors have already suggest to use high-speed and DA convertor, it might not be enhanced significantly. Is it possible to use much faster scanning galvanometer in this SCM?

Reviewer #2 (Remarks to the Author):

In this paper, the authors have proposed information entropy to evaluate the disorder of an optimized planar lens. By introducing the concept of information entropy, they have predicted that an objective chip with its entropy at $S_0=0.5$ can maintain the imaging and super-focusing properties simultaneously. This is a significant step forward to realize planar objectives with subdiffraction-limit resolution in both focusing and imaging. I believe this is an interesting breakthrough that deserves to be published in Nature Communication. The paper is well written. Analysis and discussions are comprehensive and repeatable benefiting from detailed supplementary information. However, there are several minor concerns that should be addressed before publication.

- First issue is to call it an optimized planar lens. Metasurface lenses (metalenses) are also planar lenses. And the authors have confirmed that metalenses have better efficiency and performance (Line 237: Although our achieved efficiency is not as high as those of traditional objectives and metalenses, it still exhibits significant enhancement in comparison with those of amplitude-type planar diffractive lenses). So it is not fair to call it "an optimized planar lens" which would mean among all kind of planar lenses.

- Efficiency is an important matter, however only one experimental point has been demonstrated in Figure 2e. Can more experimental data points be provided to compare the efficiency in this Figure??

- For a fair comparison, can the authors include the following information to the relevant figures:
>Working distance and magnifications of Coherent BF and Traditional RSCM in Figures 3d, e, f, and Figure 4 b,c,d
>Magnifications for all cases in Supplementary Table 1.

-Somewhere in the paper, Pls include what RSCM stands for (= reflective scanning confocal microscopies).

- Some of the figures, for example, the insets in Figures 1a and 2a are not readable. Please consider rearranging them.

REVIEWER COMMENTS

Reviewer #1 (Remarks to the Author):

The authors have demonstrated a planar diffractive lens that has both super-focusing and wide-field capabilities, which are challenging to be realized in a single planar device. To solve the problem of realizing super-focusing and wide-field issues, they introduce the concept of information entropy to evaluate the disorder of such optimized structures, which generally destroy the wide-field imaging. By using the theory of entropy change, they have suggested a special point $S=0.5$, which is considered as a good point of balancing the super-focusing and wide-field imaging. Thus, they propose a theoretical criterion to realize the bifunctional capabilities. By using the detailed optimization, they design and fabricate the expected lens with the entropy $S=0.53$, which is close to the ideal point $S=0.5$. They have also verified the super-focusing and wide-field imaging in experiment, which has good agreement with the simulations. It verifies the feasibility of their proposed method. Then, the authors have also demonstrated the planar-lens-based scanning confocal microscopy with sub-diffraction-limit resolution, which removes the usage of traditional objective lens and therefore is superior to all the reported planar lenses. So, the novelty of this manuscript is high and the experimental and theoretical results are solid for publications in Nature Communications. I might recommend its publications in Nat. Commun. if the following issues are clarified.

Reply: We thank the reviewer for the insightful comments and the strong support to our work.

1. The proposed objective chip is based on the binary-phase FZP. If the objective chip is based on the pure-phase metalenses, how can one implement the design and optimization of such bifunctional objective chips?

Reply: The design strategy for a pure phase metalens is nearly the same as that of our proposed method. The objective chip can be taken as a metalens with an additional binary-phase mask. The binary-phase mask is the same, the only difference is that the zone plates with phase modulation of 0 or π in our objective chip are substituted by the discrete metasurface nanostructures. All the design processes in our manuscript can be directly used to develop such metalens-based objective chip.

Note that, optical properties of the metalens-based objective chip is highly dependent on the nanostructures, which have two effects on focusing and imaging, respectively:

For sub-diffraction-limit focusing, the metalenses could have a multilevel phase distribution, which is similar with a multi-step FZP. So, the pure-phase metalenses could increase optical efficiency of the objective chip as compared to our current binary phase one.

For imaging, unlike multilevel phase FZP, the phase of metalenses is realized by meta-atoms with the required local phase such as propagation phase or geometric phase.

These meta-atoms are normally optimized to maintain high efficiency only under normal incidence. It will lead to difficulties in direct wide-field imaging for high-NA metalenses due to the low conversion efficiency of light with high spatial frequencies illuminating the object. Thus, the imaging performance of metalens-based objective chip is dependent on the meta-atoms, which should be designed to exhibit high efficiency under oblique illumination.

2. In the design of objective chips, the authors have used the 5-ring phase mask. Is it possible to discuss whether it is possible by using the phase mask with different (4 or even smaller, 6 or even larger) rings?

Reply: It is possible to realize objective chips by using phase mask with different number of rings. As shown in Eq. S1(**Supplementary Section1**), the total electric field at the focal plane of objective chips is tightly dependent on the number and positions of **phase-reversed zones** contained in the even rings of N-ring phase mask, if the odd and even rings have the phase of 0 and π , respectively. Because the phase-reversed zones are the key parts that determine the entropy of the proposed lens, changing the number of the few-ring phase mask will not directly affect the entropy of the objective chip. Thus, it is possible in theory to demonstrate our objective chip with less or more rings in the phase mask.

But, we should emphasize that the larger rings in the phase mask mean the more unknown parameters to be optimized, which increases the complexity of lens design. And the fewer rings will reduce the freedom of degree of design. The 5-ring phase mask in this work is a balance between lens performance and design complexity.

3. The fabricated objective chip has the deviation of 75 nm, which is small compared with the total size of the chip. But, such a deviation is large enough if it is compared with the local width of the outermost rings. So, it is better to provide more data or discussions about such fabrication errors.

Reply: As mentioned in main text, the deviation of 75nm of zones width may “*comes from insufficient exposure time of photoresist under DUV radiation*”. To evaluate the influence of insufficient etching width, we simulate the intensity profiles near the focal plane of objective chips with a deviation of 75nm by increasing and decreasing the design radius of even and odd rings by 37.5nm respectively. As shown in Fig. R1(a) and (b), the insufficient etching width has no significant influence on the focus profile except the reduced peak intensity, *i.e.*, decreasing optical focusing efficiency.

We add a short discussion in the end of section “**Theoretical efficiency of the objective chip**” in **Method**, which refers to “*In fact, due to the insufficient etching width, the focusing efficiency will further decrease, which is the reason why measured efficiency is lower than theoretical efficiency(Fig.2f).*”

Fig.R1 The simulated focal (a) and on-axis (b) intensity profiles of objective chips with (red line) and without (black line, ideal lens) insufficient etching widths.

4. The authors have suggested that the focusing efficiency of objective chip can be even enhanced by using high-efficiency dielectric metasurfaces. Indeed, it could increase the focusing efficiency, but the imaging efficiency decreases because the large-angle rays from the objects have small conversion efficiency (as pointed out in the introduction part). So, it is better to suggest the efficient method to enhance the efficiency of both focusing and imaging. Or removing the suggestion of “high-efficiency dielectric metasurfaces”.

Reply: Thank the reviewer for the valuable suggestion. We agree that the metalens might not be the best candidate to collect the light with large-angle rays. As suggested, we have deleted the relevant claim about “*high-efficiency dielectric metasurfaces*” in our revised manuscript.

5. The scanning speed is slow now (100*100 pixels takes 15 minutes). Although the authors have already suggest to use high-speed and DA convertor, it might not be enhanced significantly. Is it possible to use much faster scanning galvanometer in this SCM?

Reply: Thanks for the suggestion about scanning galvanometer that indeed offers fast angular scanning. One issue of using galvanometer is that the objective chip must work under tilting illumination. At the current stage, our designed objective chips are not optimized for the case of oblique incidence. Thus, the focus will deform under the tilting incidence, which will destroy the imaging quality of objective-chips-based reflection scanning confocal microscope. On the other hand, if the coma of the objective chip is removed, the scanning galvanometer will significantly enhance the scanning speed, which will be addressed in our future work.

Reviewer #2 (Remarks to the Author):

In this paper, the authors have proposed information entropy to evaluate the disorder of an optimized planar lens. By introducing the concept of information entropy, they have predicted that an objective chip with its entropy at $S_0=0.5$ can maintain the imaging and super-focusing properties simultaneously. This is a significant step forward to realize planar objectives with subdiffraction-limit resolution in both focusing and imaging. I believe this is an interesting breakthrough that deserves to be published in Nature Communication. The paper is well written. Analysis and discussions are comprehensive and repeatable benefiting from detailed supplementary information. However, there are several minor concerns that should be addressed before publication.

Reply: We greatly appreciate the reviewer for appreciating the impact of our work and acknowledging the quality of our data and presentation.

- First issue is to call it an optimized planar lens. Metasurface lenses (metalenses) are also planar lenses. And the authors have confirmed that metalenses have better efficiency and performance (Line 237: Although our achieved efficiency is not as high as those of traditional objectives and metalenses, it still exhibits significant enhancement in comparison with those of amplitude-type planar diffractive lenses). So it is not fair to call it "an optimized planar lens" which would mean among all kind of planar lenses.

Reply: Thank the reviewer for pointing out this important issue. To avoid any confusion, we have modified the claim in our revised manuscript by using "Although our achieved efficiency is not as high as those of traditional objectives and metalenses, it still exhibits significant enhancement in comparison with those of amplitude-type *zone-plate* lenses".

For the issue "an optimized planar lens", our strategy of defining the entropy is valid for any planar lens, including both zone-plate lens and metalens. In fact, our objective chip could be made by using a metalens and an additional few-ring phase mask, please see our reply to comment 1 from the referee 1. To make it more rigorous, we amend it as "*an optimized zone-plate lens*", which has been updated in the **Conclusion** part of our revised manuscript.

2. Efficiency is an important matter, however only one experimental point has been demonstrated in Figure 2e. Can more experimental data points be provided to compare the efficiency in this Figure??

Reply: To provide more experimental data, we fabricate two other objective chips with 350 nm and 390 nm etched depths, which yield the phase modulation of 0.81π and 0.905π , respectively. By using the same measurement method, the experimental focusing efficiency are shown in Fig. 2f of the revised manuscript, exhibiting the expected shape as the simulated ones. For better reading, we have pasted Fig. 2f in the following:

3. For a fair comparison, can the authors include the following information to the relevant figures:

>Working distance and magnifications of Coherent BF and Traditional RSCM in Figures 3d, e, f, and Figure 4 b,c,d

>Magnifications for all cases in Supplementary Table 1.

Reply: The working distance and magnification of objective used in coherent BF microscope and traditional RSCM(Fig. 3d, e and Fig. 4b, c) are 310 μ m and 100 \times , respectively. We add this information to the captions of Fig. 3 and 4, as “*The working distance and magnification of the objective used in coherent BF microscope and traditional RSCM is 310 μ m and 100 \times , respectively. For a better comparison, the pictures in (d) are scaled down by its magnification of 255.3 \times .*” and “*In (b-c), all the parameters (i.e.,the working distance and magnification of objectives used in coherent BF microscope and traditional RSCM) are the same as those in Fig. 3d, e.*”, respectively. The images shown in the Fig. 3f and Fig. 4d are the results of our objective-chips-based RSCM, the working distance of our objective chip is 1mm.

In addition, according to the definition in geometric optics, the image magnification is the ratio of image size to object size. The SCM images in this work are the direct results of point-to-point scanning without any magnification. Therefore, the magnifications of all SCM images shown in Fig. 3e, f, Fig. 4c, d and all cases in Supplementary Table 1 are 1. In the caption of Supplementary Table 1, we provide a brief instruction “*Note: The image magnifications of all these SCMs in Supplementary Table 1 are 1 \times .*”.

For coherent BF microscope (Fig. 3d and Fig. 4b), the image magnification can be calculated directly. The image length of slits covers 148 pixels and each pixel size of CCD camera is 3.45 μ m, which leads to the image length of 510.6 μ m. On the other hand, the actual length of slits is 2 μ m. Therefore, the image magnification of coherent BF microscope (Fig.3d) is $M = \frac{510.6\mu m}{2\mu m} = 255.3$. The coherent BF microscope used

in Fig. 3d is same as that of Fig. 4b. Hence, the image magnification of Fig.3d and Fig.4b are both $M=255.3$. For a better comparison, the images (Fig.3d and Fig.4b) of coherent BF microscope are scaled down by the magnification $M=255.3$. We add the brief instructions into the captions of Fig.3d and 4b as “*For a better comparison, the pictures in (d) are scaled down by its magnification of 255.3 \times .*”, thus clarifying this issue in

our revised manuscript.

4. Somewhere in the paper, Pls include what RSCM stands for (= reflective scanning confocal microscopies).

Reply: Thank the referee for pointing out this typo. The RSCM stands for reflective scanning confocal microscopies. We provide its full name of RSCM when it appears first time in **Introduction** of main text, as shown by “Without involving these issues, *reflective SCMs (RSCM)* are, therefore, more popular for noninvasive and in vivo imaging of various specimens”.

5. Some of the figures, for example, the insets in Figures 1a and 2a are not readable. Please consider rearranging them.

Reply: In our revised manuscript, we rearrange the insets of Fig. 1a and Fig. 2a. For your better reading, we copy them in the following:

Fig. 1

Fig. 2

Reviewer #1 (Remarks to the Author):

Since the authors have addressed all my concerns, I therefore recommend the publication of this manuscript.

Reviewer #2 (Remarks to the Author):

The authors have addressed my concerns. I am now in a position to recommend this paper for publication in Nature Communications.

REVIEWERS' COMMENTS AND REPLIES

Reviewer #1 (Remarks to the Author):

Since the authors have addressed all my concerns, I therefore recommend the publication of this manuscript.

Reply: We thank the referee for his/her strong recommendation and efforts to our work.

Reviewer #2 (Remarks to the Author):

The authors have addressed my concerns. I am now in a position to recommend this paper for publication in Nature Communications.

Reply: We thank the referee for the strong support to our work.